# Lipocalin, Resistin and Gut Microbiota-Derived Propionate Could Be Used to Predict Metabolic Bariatric Surgery Selected Outcomes

Teresa Auguet [1,2,*], Miguel Lopez-Dupla [1], Jessica Ramos [1], Laia Bertran [2], David Riesco [1,2], Carmen Aguilar [2], Anna Ardévol [3], Montserrat Pinent [3], Fàtima Sabench [2,4], Daniel Del Castillo [2,4] and Cristóbal Richart [2,*]

1    Internal Medicine Unit, Hospital Universitari Joan XXIII, 43007 Tarragona, Spain;
     jmlopezdupla.hj23.ics@gencat.cat (M.L.-D.); jramos.hj23.ics@gencat.cat (J.R.);
     driesco.hj23.ics@gencat.cat (D.R.)
2    GEMMAIR Research Group (AGAUR)-Medicina Aplicada, Department of Medicine and Surgery,
     Institut d'Investigació Sanitària Pere Virgili (IISPV), Universitat Rovira i Virgili (URV), 43007 Tarragona, Spain;
     laia.bertran@urv.cat (L.B.); caguilar.hj23.ics@gencat.cat (C.A.); fatima.sabench@urv.cat (F.S.);
     danieldel.castillo@urv.cat (D.D.C.)
3    MoBioFood Research Group, Department of Biochemistry and Biotechnology, Universitat Rovira i Virgili,
     43007 Tarragona, Spain; anna.ardevol@urv.cat (A.A.); montserrat.pinent@urv.cat (M.P.)
4    Surgery Unit, Hospital Sant Joan de Reus, Department of Medicine and Surgery, IISPV, URV,
     43204 Reus, Spain
*    Correspondence: tauguet.hj23.ics@gencat.cat (T.A.); cristobalmanuel.richart@urv.cat (C.R.);
     Tel.: +34-977-295-833 (T.A.)

**Abstract:** Many patients with clinically severe obesity (CSO) need to undergo bariatric surgery, with possible side effects, so individualized predictive methods are required. Adipocytokines and gut/intestinal microbiota-derived metabolites could be predictive biomarkers of metabolic success post-surgery, but the knowledge in this field is undefined. The objective of this work was to determine whether adipocytokines and microbiota-derived metabolites can be used to predict the metabolic improvement post-surgery in women with CSO. We analyzed circulating levels of some cytokines and some microbiota-derived metabolites at baseline and 12 months post-surgery from 44 women with CSO and 21 women with normal weight. Results showed that glucose, insulin, glycosylated hemoglobin A1c (HbA1c), low-density lipoprotein (LDL-C), and triglycerides levels were decreased post-surgery, while high density lipoprotein increased. Twelve months later, leptin, resistin, lipocalin, PAI-1, TNF-α, and IL-1β levels were lower than baseline, meanwhile adiponectin, IL-8, and IL-10 levels were increased. Moreover, baseline lipocalin levels were associated with HbA1c reduction post-surgery; meanwhile baseline resistin was related to postoperative HOMA2 (insulin resistance) and baseline propionate was associated with LDL-C decrease. To conclude, the detection of lipocalin, resistin, and propionate levels may be used to predict the metabolic success following bariatric surgery, although new knowledge is needed.

**Keywords:** adipocytokines; microbiota; obesity; bariatric surgery; biomarker

## 1. Introduction

Obesity is the epidemic of the 21st century. The hormonal disorder and the pro-inflammatory effects of the excessive adipose tissue are associated with an increased risk of cancer and also some serious non-neoplastic conditions such as metabolic syndrome, type 2 diabetes mellitus (T2DM), and cardiovascular disease (CVD) [1].

Bariatric surgery is an efficacious therapeutic method for clinically severe obesity (CSO), leading to marked weight loss and the improvement of linked comorbidities, such as CVD and T2DM, when low-calorie diet, physical exercise, and behavioral and drug

therapy fail [2–4]. Sleeve gastrectomy (SG), a primarily restrictive technique, is the bariatric procedure most commonly performed due to its easiness and low rate of complications, and it represents 53.6% of all bariatric surgeries, followed by Roux-en-Y gastric bypass (RYGB) that represents 30.1% [5]. While RYGB and SG are effective in the improvement of metabolic complications, they involve different mechanisms, which are largely unknown [6]. Longitudinal studies showed that bariatric surgery is also useful for diseases including T2DM and CVD [7], but not without potential adverse effects such as metabolic bone disease, gallstone development or iron, vitamin B12, fat-soluble vitamins, thiamine, and folate deficiency [8,9]. Moreover, in patients with CSO and T2DM with poor metabolic control, bariatric surgery is indicated to treat metabolic comorbidities [10].

Although bariatric surgery provides adequate and sustainable weight loss and T2DM remission, 15–20% of subjects do not achieve these targets [3,11]. Given that a large number of people need this procedure, which is not without potential side effects, it would be important to choose individually the best surgical technique, to obtain the best metabolic response, using predictive biomarkers. Regarding T2DM, several remission predictors were found like age, pathology duration, baseline C-peptide levels, glycosylated hemoglobin A1c (HbA1c), or previous insulin treatment [12–14]. Additionally, some predictive outcome scores for T2DM remission have been developed with contradictory results [15]. Therefore, there is a necessity to conceive personalized techniques to predict the degree of metabolic comorbidity reversal after bariatric surgery.

It is known that a link exists between inflammation and obesity, as the excessive adipose tissue in subjects with obesity tends to initiate the immune process in white adipose tissue (WAT), liver, and immune cells [16]. This excessive mass of adipose tissue promotes the secretion and release of pro-inflammatory cytokines, such as interleukin (IL)-1β, IL-6, tumor necrosis factor-α (TNF-α), and leptin, and stimulates monocyte chemo attractant protein-1 (MCP-1), which consequently reduces the generation of adiponectin thereafter giving place to a pro-inflammatory phase [17]. Not only inflammation but also intestinal dysbiosis is linked to insulin resistance (IR) and obesity. Lipopolysaccharides (LPS) from intestinal microbiota can contribute to chronic subclinical inflammatory processes in obesity, leading to IR through activation of Toll-like receptor (TLR)-4 [18]. Reduced levels of circulating short-chain fatty acids (SCFAs) can substantially intervene in insulin sensitivity reduction and obesity [18]. Other mechanisms include the effects of bile acids, branched-chain amino acids (BCAAs), and other less known factors [18].

Since many patients with CSO need to undergo bariatric or metabolic surgery, with possible side effects, individualized predictive methods of metabolic response are required. With all these antecedents, the purpose of this work is to study preoperative and postoperative circulating cytokines and their involvement in any changes in metabolic factors due to bariatric surgery in a cohort of women with CSO. Additionally, we wanted to evaluate the predictive value of circulating gut microbiota-related metabolites in metabolic improvement after bariatric surgery.

## 2. Materials and Methods

### 2.1. Subjects

In the present work, we analyzed the interleukin and adipocytokine circulating levels in 44 Caucasian women with CSO (body mass index, BMI > 35 kg/m$^2$), and 21 normal weight (NW) women with a BMI < 25 kg/m$^2$. Our cohort is made up of only women because the most patients who undergo bariatric surgery are women. Additionally, we wanted to perform this study in a homogenous group in order to avoid the interference of some confounding factors like age or gender. It is well-known that the body composition differs between men and women, and also the energy imbalance and hormones. Moreover, there are sex-specific differences in lipid and glucose metabolism [19]. Follow-up samples were collected only from women with CSO who underwent a laparoscopic bariatric surgery given that we have made comparison between baseline levels of control and CSO subjects, and between levels of CSO at baseline and at 12 months postoperatively. We have not made

a follow-up of the patient controls since these have not undergone any metabolic change and their levels should be comparable with the basal ones. There was no loss to follow-up in CSO patients. Women with CSO were evaluated at two time-points: at the moment of laparoscopic bariatric surgery (baseline), and at 12 months later than the surgery. A work flow of the population studied was represented in Figure 1.

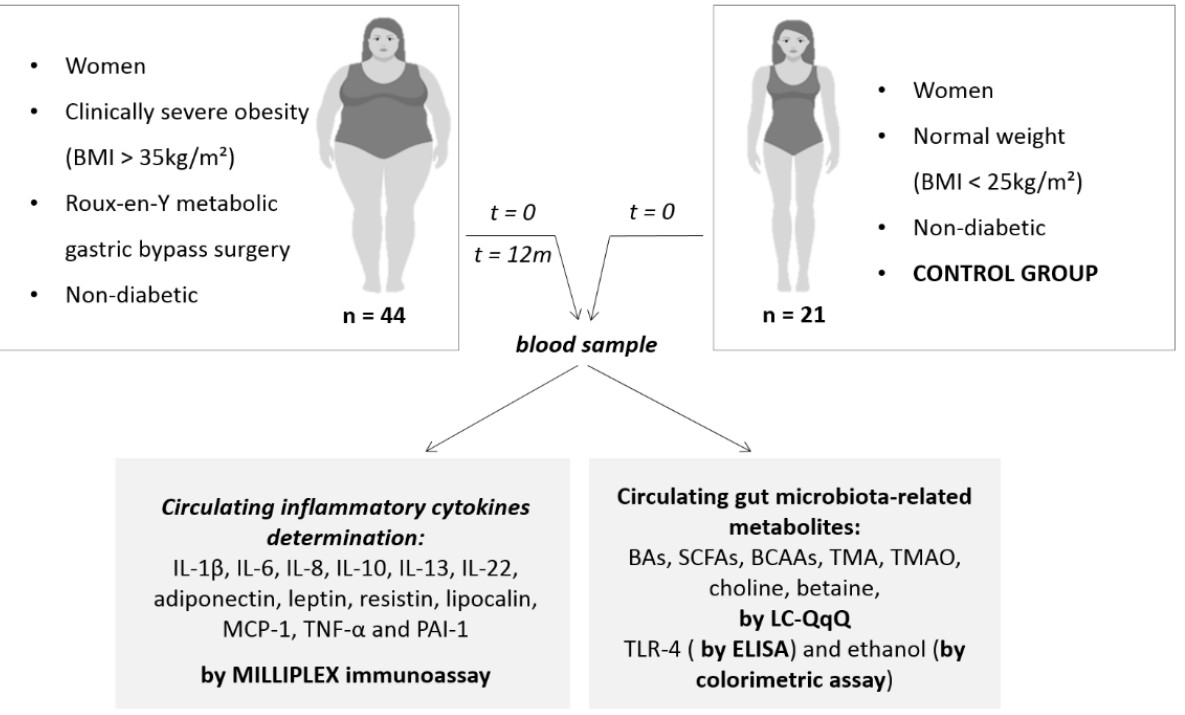

**Figure 1.** We analyzed circulating inflammatory cytokine levels and levels of circulating gut microbiota-related metabolites in blood of women with severe obesity at the moment of the bariatric surgery and 12 months after; and in a control group of women with normal weight. BMI, body mass index; n, count; t, time; m, months; IL, interleukin; MCP-1, monocyte chemo attractant protein 1; TNF-α, tumor necrosis factor alpha; PAI-1, plasminogen activator inhibitor 1; BAs, bile acids; SCFAs, short chain fatty acids; BCAAs, branched-chain amino acids; TMA, trimethylamine; TMAO, trimethylamine-N-oxide; LC-QqQ, triple-quadrupole-mass spectrometry; TLR-4, Toll-like receptor 4; ELISA, enzyme-linked immunosorbent assay.

This work was approved by the Institutional Review Board (CEIm: 161C/2016), and all participants gave written informed consent.

The CSO patients received a very low-calorie diet for the last three months prior to the surgery. Patients who presented an acute disease, acute or chronic inflammatory or infective illnesses or end state malignant neoplasia were excluded. Post-menopausal women and women that receive contraceptive treatment were excluded too. Although patients who underwent a bariatric surgery have the commitment not to get pregnant for two years, not all the patients take hormonal contraceptive methods. Given that hormonal contraceptive treatment uses to have an impact on the lipid metabolism [20] we excluded the patients who took it to avoid this bias.

### 2.2. Anthropometrical and Biochemical Analysis

Anthropometrical analysis included the measurement of weight, height, and BMI calculation. Plasma samples, which were obtained from women with CSO, were stored at −80 °C. Laboratory analysis incorporated glucose, insulin, lower high density lipoprotein-cholesterol (HDL-C), low density lipoprotein-cholesterol (LDL-C), HbA1c, and triglycerides,

which were carried out using an automated analyzer and measured on an empty stomach. IR was estimated using homeostasis model assessment 2 of insulin resistance (HOMA2-IR).

### 2.3. Plasma Measurements

We analyzed serum levels of different inflammatory cytokines such as IL-1β, IL-6, IL-8, IL-10, IL-13, and IL-22, and other adipocytokines including adiponectin, leptin, resistin, lipocalin, MCP-1, TNF-α, and also plasminogen activator inhibitor-1 (PAI-1). Circulating levels of IL-1β, IL-6, IL-8, IL-10, IL-13, IL-22, and TNF-α were determined using multiplex sandwich immunoassays and the MILLIPLEX MAP Human High Sensitivity T Cell Magnetic Bead Panel (HSTCMAG-28SK-07, Millipore, Billerica, MA, USA). Circulating levels of adiponectin, leptin, resistin, lipocalin, PAI-1, and MCP-1 were measured using MILLIPLEX MAP Human Adipokine Magnetic Bead Panel 1 (HADK1MAG-61K-04, Millipore, Billerica, MA, USA) and MILLIPLEX MAP Human Adipokine Magnetic Bead Panel 2 (HADK2MAG-61K-02, Millipore, Billerica, MA, USA). The whole assay was assessed using the Bio-Plex 200 instrument, according to the manufacturer's instructions. TLR-4 levels were determined using enzyme-linked immunosorbent assay (ELISA) according to the manufacturer procedure (Ref. SEA753Hu; USCN, Wuhan, China).

On the one hand, 15 BAs were absolutely quantified (CA, cholic acid; CDCA, chenodeoxycholic acid; DCA, deoxycholic acid; GCA, glycocholic acid; GCDCA, glycochenodeoxycholic acid; GDCA, glycodeoxycholic acid; LCA, lithocholic acid; TCA, taurocholic acid; TCDCA, taurochenodeoxycholic acid; TDCA, taurodeoxycholic acid; TLCA, taurolithocholic acid; TUDCA, tauroursodeoxycholic acid and UDCA, ursodeoxycholic acid); on the other hand, 2 BAs were relatively quantified (GLCA, glycolithocholic acid; GUDCA, glycoursodeoxycholic acid) by liquid chromatography coupled to triple-quadrupole-mass spectrometry (LC-QqQ). Choline, trimethylamine (TMA), trimethylamine *N*-oxide (TMAO), betaine, SCFAs (acetic, butyric and propionic acid), and BCAAs (isobutyric and isovaleric acid) plasma samples were also absolutely quantified by LC-QqQ.

### 2.4. Statistical Analysis

Provided data were expressed as mean and standard deviation (SD). Variables with a non-parametric distribution were converted to logarithms to use parametric analytical methods using the IBM SPSS Statistics v23 for Windows (Chicago, IL, USA). Differences between the CSO group and the NW controls were determined through Student's *t* test (independent or related samples) for parametrically distributed variables. The strength of correlations between variables was evaluated using Pearson's method. Multiple linear regression analysis with backward variable selection was assessed to find independent predictors of BMI, HbA1c, and triglycerides reduction, and HDL-C increase after surgical procedure. The validity of the regression model and its assumptions were carried out with the plot of residuals versus predicted. *p* values $< 0.05$ were considered to be statistically significant. Graphics were created using GraphPad Prism v7 program (San Diego, CA, USA) and Metaboanalyst software (Québec, QC, Canada).

## 3. Results

### 3.1. Characteristics of Patients and Adipocytokine Levels at Baseline and Postsurgery

Baseline characteristics of patients, biochemical variables, and cytokine concentrations are expressed in Table 1. Biochemical analyses indicated that CSO patients presented increased weight, BMI, glucose, insulin, HOMA2-IR, triglycerides, aspartate aminotransferase (AST), alanine aminotransferase (ALT), gamma-glutamyltransferase (GGT), alkaline phosphatase (ALP), and C-peptide levels than the controls did. The analyses also revealed that CSO patients had decreased levels of HDL-C than the controls. Leptin, PAI-1, MCP-1, resistin, lipocalin, and IL-10 levels were higher in CSO group versus controls (Figure 2).

**Table 1.** Anthropometrical parameters, biochemical variables, and cytokine levels of the whole study cohort.

| Variables | NW (*n* = 21) | CSO (*n* = 44) | *p*-Value |
|---|---|---|---|
| Age (years) | 44.48 ± 8.69 | 49.02 ± 8.18 | 0.060 |
| Weight (kg) | 57.56 ± 7.01 | 112.13 ± 13.27 | <0.001 * |
| BMI (kg/m$^2$) | 21.97 ± 2.03 | 42.89 ± 5.14 | <0.001 * |
| Glucose (Log10 mg/dL) | 1.90 ± 0.04 | 1.97 ± 0.14 | 0.006 * |
| Insulin (Log10 mU/L) | 0.76 ± 0.13 | 0.98 ± 0.33 | 0.003 * |
| A1C (Log10 %) | 0.73 ± 0.03 | 0.76 ± 0.07 | 0.222 |
| HOMA2-IR (Log10) | −0.14 ± 0.13 | 0.10 ± 0.33 | 0.002 * |
| Cholesterol (Log10 mg/dL) | 2.26 ± 0.08 | 2.24 ± 0.10 | 0.526 |
| HDL-C (Log10 mg/dL) | 1.84 ± 0.09 | 1.64 ± 0.10 | <0.001 * |
| LDL-C (Log10 mg/dL) | 1.98 ± 0.13 | 2.01 ± 0.13 | 0.463 |
| Triglycerides (Log10 mg/dL) | 1.80 ± 0.19 | 2.06 ± 0.21 | <0.001 * |
| AST (Log10 UI/L) | 1.27 ± 0.10 | 1.41 ± 0.25 | 0.002 * |
| ALT (Log10 UI/L) | 1.22 ± 0.16 | 1.44 ± 0.25 | 0.001 * |
| GGT (Log10 UI/L) | 1.14 ± 0.14 | 1.40 ± 0.31 | <0.001 * |
| ALP (Log10 UI/L) | 1.75 ± 0.10 | 1.82 ± 0.09 | 0.012 * |
| C-Peptide (Log10 ng/mL) | 0.05 ± 0.11 | 0.27 ± 0.20 | 0.002 * |

Data were expressed as the mean ± standard deviation (SD) of the log10 of each variable. Differences between groups were evaluated using the Student's *t* test. * *p* < 0.05 is considered significant. NW, normal weight control (BMI < 25 kg/m$^2$); CSO, clinically severe obesity (BMI > 35 kg/m$^2$). BMI, body mass index; A1C, glycosylated hemoglobin A1c; HOMA2-IR, the homeostasis model assessment 2 of insulin resistance; HDL-C, high density lipoprotein-cholesterol; LDL-C, low density lipoprotein-cholesterol; AST, aspartate aminotransferase; ALT, alanine aminotransferase, GGT, gamma-glutamyltransferase; ALP, alkaline phosphatase.

Then, in the CSO patients, we studied anthropometrical characteristics, metabolic variables, and cytokine concentrations at baseline, and 12 months later (Table 2). At the surgery moment, weight, BMI, fasting glucose, insulin, HbA1c levels, and HOMA2-IR were markedly decreased. HDL-C levels increased, while low density lipoprotein-cholesterol (LDL-C), C-peptide, and triglyceride levels were much lower, postoperatively. Moreover, systolic blood pressure was markedly reduced after surgery. Additionally, AST, ALT, GGT, and ALP levels were reduced post-surgery.

**Table 2.** Anthropometrical parameters and metabolic variables of the cohort group with SO, at baseline and 12 months post-operatively.

| Variables | Baseline | Follow-Up | *p*-Value |
|---|---|---|---|
| Weight (kg) | 112.13 ± 13.27 | 78.54 ± 14.13 | <0.001 * |
| BMI (kg/m$^2$) | 42.89 ± 5.14 | 30.04 ± 5.34 | <0.001 * |
| Glucose (Log10 mg/dL) | 1.97 ± 0.14 | 1.91 ± 0.05 | 0.003 * |
| Insulin (Log10 mU/L) | 0.96 ± 0.29 | 0.55 ± 0.27 | <0.001 * |
| HbA1C (Log10 %) | 0.76 ± 0.07 | 0.72 ± 0.05 | 0.002 * |
| HOMA2-IR (Log10) | 0.07 ± 0.29 | −0.33 ± 0.27 | <0.001 * |
| Cholesterol (Log10 mg/dL) | 2.24 ± 0.10 | 2.22 ± 0.06 | 0.139 |
| HDL-C (Log10 mg/dL) | 1.64 ± 0.10 | 1.78 ± 0.08 | <0.001 * |
| LDL-C (Log10 mg/dL) | 2.01 ± 0.13 | 1.93 ± 0.12 | <0.001 * |
| Triglycerides (Log10 mg/dL) | 2.06 ± 0.21 | 1.90 ± 0.14 | <0.001 * |
| AST (Log10 UI/L) | 1.41 ± 0.25 | 1.29 ± 0.12 | 0.002 * |
| ALT (Log10 UI/L) | 1.43 ± 0.25 | 1.25 ± 0.20 | <0.001 * |
| GGT (Log10 UI/L) | 1.40 ± 0.31 | 1.06 ± 0.21 | <0.001 * |
| ALP (Log10 UI/L) | 1.82 ± 0.09 | 1.93 ± 0.09 | <0.001 * |
| C-Peptide (Log10 ng/mL) | 0.24 ± 0.15 | 0.07 ± 0.21 | 0.001 * |

Data were represented as the mean ± standard deviation (SD) of the log10 of the variable. Variances between groups were evaluated using the Student's *t* test. * *p* < 0.05 is considered statistically relevant. BMI, body mass index; A1C, glycosylated hemoglobin A1c; HOMA2-IR, the homeostasis model assessment 2 of insulin resistance; HDL-C, high density lipoprotein-cholesterol; LDL-C, low density lipoprotein-cholesterol; AST, aspartate aminotransferase; ALT, alanine aminotransferase, GGT, gamma-glutamyltransferase; ALP, alkaline phosphatase.

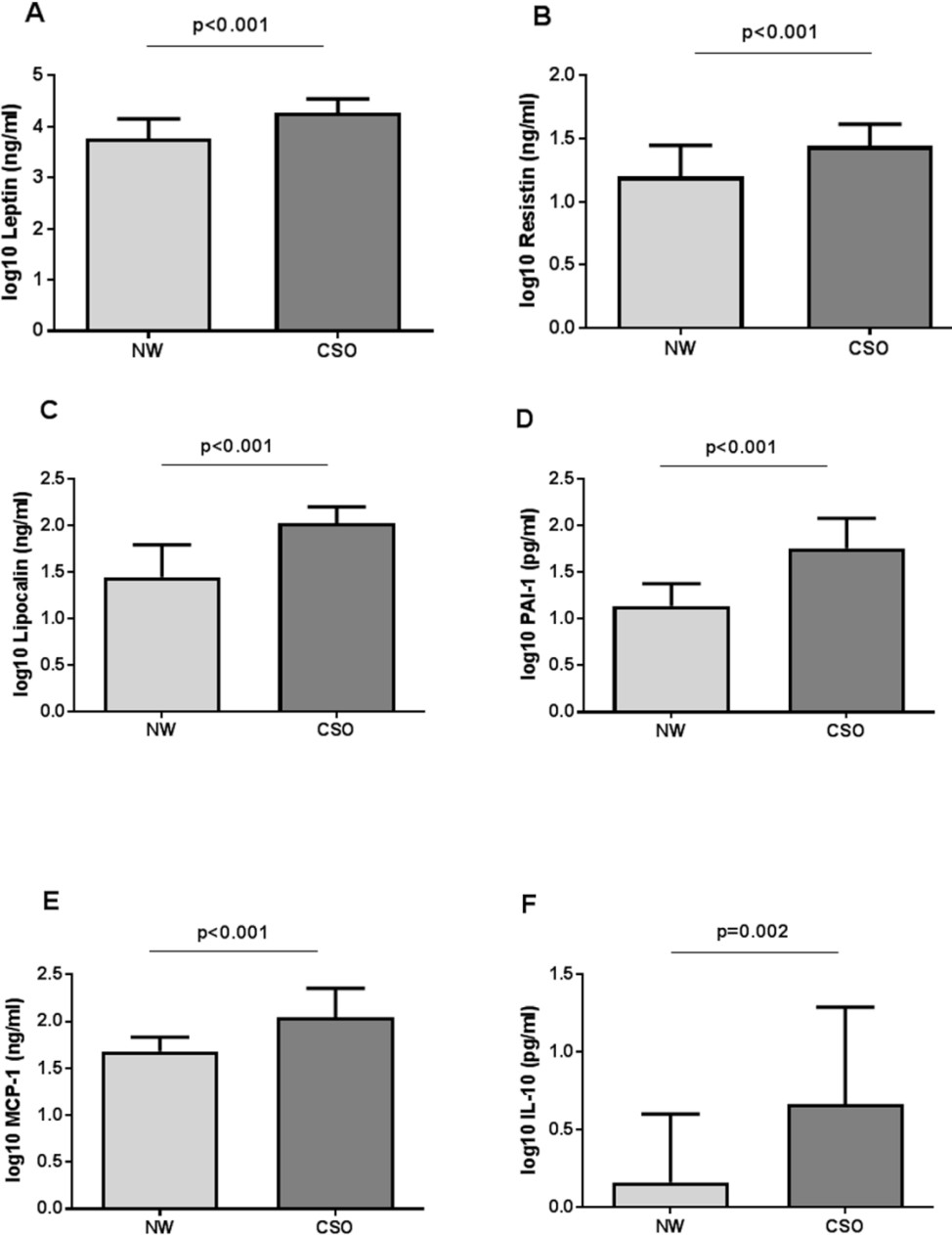

**Figure 2.** Relevant changes of (**A**) Leptin, (**B**) Resistin, (**C**) Lipocalin, (**D**) PAI-1, (**E**) MCP-1 and (**F**) IL-10 circulating levels between normal weight (NW) women and women with clinically severe obesity (CSO). Differences between groups were calculated using the Student's *t*-test. Differences between NW and CSO group were statistically significant when $p < 0.05$. PAI-1, plasminogen activator inhibitor-1; MCP-1, monocyte chemo attractant protein 1; IL, interleukin.

Regarding adipocytokines, leptin, PAI-1, resistin, lipocalin, TNF-α, and IL-1β levels were lower than its baseline levels, meanwhile adiponectin, IL-10 and IL-8 concentration were increased. Relevant differences are graphically represented in Figure 3. We could not find differences in MCP-1, IL-22, IL-13, and IL-6 levels, before and after bariatric surgery.

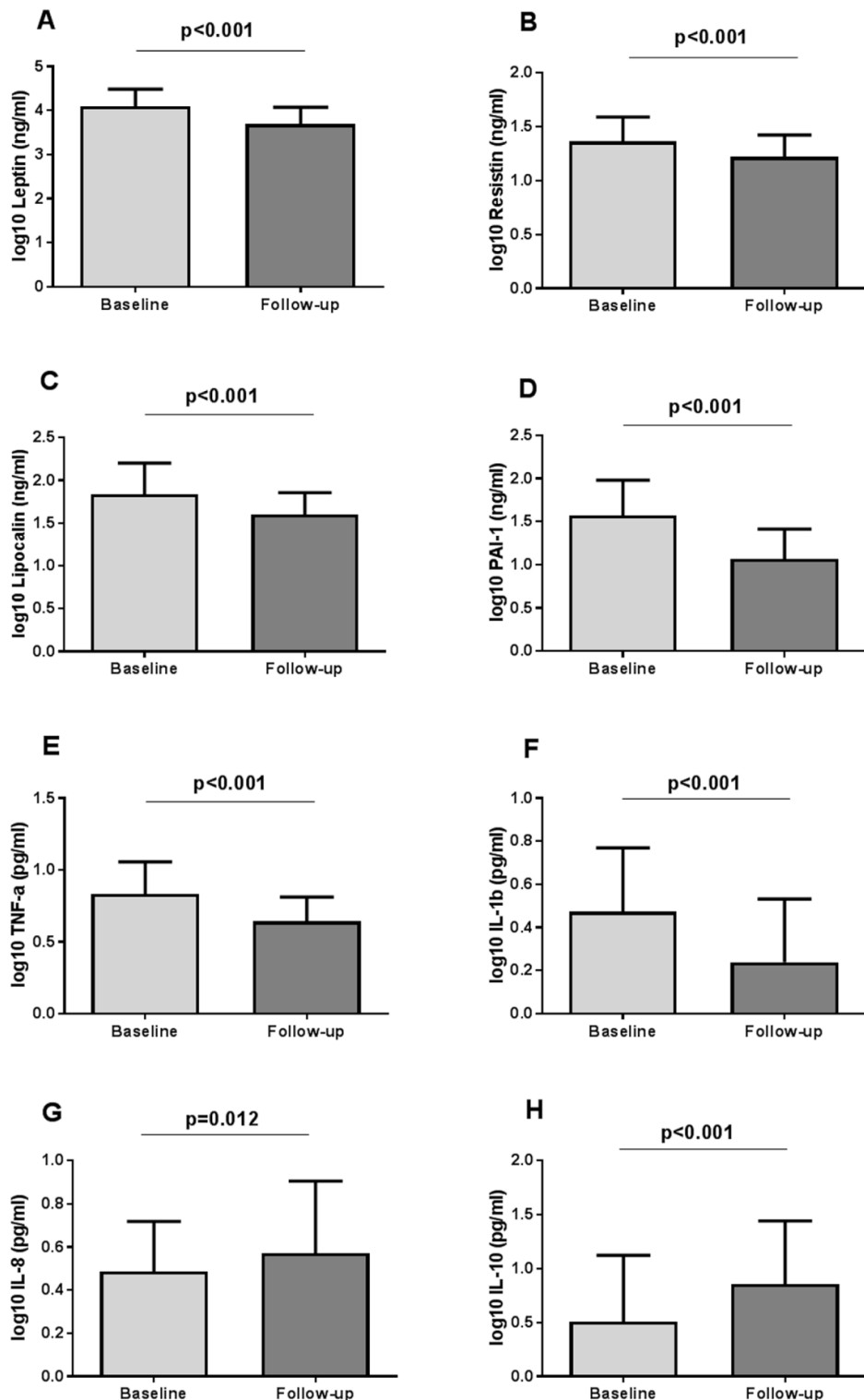

**Figure 3.** Relevant changes of (**A**) Leptin, (**B**) Resistin, (**C**) Lipocalin, (**D**) PAI-1, (**E**) TNF-α, (**F**) IL-1β, (**G**) IL-8, and (**H**) IL-10 circulating concentrations between baseline and follow-up 12 months' post-surgery in women with CSO. Differences between groups were evaluated using the Student's *t*-test for paired samples. Differences in cytokine levels between studied time-points were statistically relevant when *p* < 0.05. PAI-1, plasminogen activator inhibitor-1; TNF-α, tumor necrosis factor alpha; IL, interleukin.

### 3.2. Intestinal Microbiota-Derived Metabolites Levels in Serum Samples of the Population Studied

Later, we evaluated the serum concentrations of intestinal microbiota-derived metabolites according to obesity presence. Our results were represented in Table 3. Choline concentration was higher and betaine levels were lower in women with CSO than in controls. In the regard of SCFAs, women with CSO presented significantly lower levels of isobutyrate plasma concentration compared to NW subjects. Regarding serum levels of primary BAs, we found that CDCA and GCDCA were lower in women with CSO compared to controls. As for secondary BAs, we described reduced levels of DCA, GDCA, TLCA, TDCA, TUDCA, and GLCA in women with CSO in comparison with NW group.

**Table 3.** Serum concentrations of choline and its byproducts, betaine, soluble TLR-4, SCFAs, and primary and secondary BAs in patients with CSO and NW subjects.

| Variables | NW (*n* = 21) | CSO (*n* = 44) | *p*-Value |
|---|---|---|---|
| Choline (Log 10 µM) | 0.98 ± 0.09 | 1.23 ± 0.18 | <0.001 * |
| TMA (Log 10 nM) | 1.77 ± 0.10 | 1.65 ± 0.32 | 0.077 |
| TMAO (Log 10 µM) | 0.46 ± 0.33 | 0.46 ± 0.32 | 0.973 |
| Betaine (Log 10 µM) | 1.52 ± 0.11 | 1.42 ± 0.15 | 0.011 * |
| TLR4 (Log 10 ng/mL) | 0.44 ± 0.18 | 0.31 ± 0.34 | 0.141 |
| *Short chain fatty acids* | | | |
| Acetate (Log 10 ng/mL) | 3.20 ± 0.24 | 3.20 ± 0.38 | 0.975 |
| Propionate (Log 10 ng/mL) | 2.27 ± 0.07 | 2.31 ± 0.28 | 0.459 |
| Isobutyrate (Log 10 ng/mL) | 1.59 ± 0.80 | 1.52 ± 0.17 | 0.037 * |
| Butyrate (Log 10 ng/mL) | 1.67 ± 0.25 | 1.82 ± 0.27 | 0.053 |
| Isovalerate (Log 10 ng/mL) | 1.42 ± 0.24 | 1.29 ± 0.41 | 0.211 |
| *Primary bile acids* | | | |
| CDCA (Log 10 nM) | 2.04 ± 0.44 | 1.75 ± 0.62 | 0.047 * |
| CA (Log 10 nM) | 1.82 ± 0.60 | 1.70 ± 0.54 | 0.535 |
| GCDCA (Log 10 nM) | 2.52 ± 0.45 | 2.20 ± 0.40 | 0.012 * |
| GCA (Log 10 nM) | 1.93 ± 0.38 | 1.88 ± 0.44 | 0.705 |
| TCA (Log 10 nM) | 1.18 ± 0.33 | 1.32 ± 0.51 | 0.334 |
| TCDCA (Log 10 nM) | 1.72 ± 0.48 | 1.74 ± 0.44 | 0.869 |
| *Secondary bile acids* | | | |
| DCA (Log 10 nM) | 2.38 ± 0.52 | 1.97 ± 0.41 | 0.003 * |
| GDCA (Log 10 nM) | 2.05 ± 0.40 | 1.70 ± 0.45 | 0.006 * |
| LCA (Log 10 nM) | 1.12 ± 0.22 | 1.07 ± 0.19 | 0.460 |
| UDCA (Log 10 nM) | 1.41 ± 0.38 | 1.49 ± 0.63 | 0.656 |
| TLCA (Log 10 nM) | 0.41 ± 0.38 | 0.15 ± 0.45 | 0.045 * |
| TDCA (Log 10 nM) | 1.49 ± 0.43 | 1.07 ± 0.65 | 0.019 * |
| TUDCA (Log 10 nM) | 0.36 ± 0.30 | 0.58 ± 0.32 | 0.022 * |
| GLCA (Log 10 nM) | 1.82 ± 0.30 | 1.29 ± 0.37 | <0.001 * |
| GUDCA (Log 10 nM) | 2.49 ± 0.40 | 2.44 ± 0.51 | 0.755 |

Data were represented as the mean ± standard deviation (SD). Variances among groups were evaluated by Student's *t* test. * *p* < 0.05 was considered statistically relevant. NW, normal weight control (BMI < 25 kg/m$^2$); CSO, clinically severe obesity (BMI > 35 kg/m$^2$). TMA, trimethylamine; TMAO, trimethylamine *N*-oxide; TLR-4, Toll-like receptor-4; CDCA, chenodeoxycholic acid; CA, cholic acid; GCDCA, glycochenodeoxycholic acid; GCA, glycocholic acid; TCA, taurocholic acid; TCDCA, taurochenodeoxycholic acid; DCA, deoxycholic acid; GDCA, glycodeoxycholic acid; LCA, lithocholic acid; UDCA, ursodeoxycholic acid; TLCA, taurolithocholic acid; TDCA, taurodeoxycholic acid; TUDCA, tauroursodeoxycholic acid; GLCA, glycolithocholic acid; and GUDCA, glycoursodeoxycholic acid.

### 3.3. Correlations of Circulating Interleukin and Other Adipocytokine Levels with Anthropometrical Measures and with Metabolic Parameters

First, we analyzed the correlations between circulating interleukins with anthropometrical measures and metabolic parameters in the whole cohort (Figure 4A). To highlight, the negative correlations between PAI-1 or lipocalin and HDL-C (r = −0.392, *p* = 0.001; r = −0.236, *p* = 0.04 respectively), adiponectin and glucose (r = −0.251; *p* = 0.04) or HbA1c (r = −0.298; *p* = 0.01); and the positive associations between leptin and HOMA2-IR (r = 0.495, *p* < 0.001) or BMI (r = 384; *p* = 0.001) and PAI-1 or MCP-1 and ALT (r = 0.435, *p* < 0.001; r = 0.406, *p* < 0.001 respectively). Then, we analyzed the correlation in the CSO

cohort (Figure 4B), we emphasize the negative associations between MCP-1 and BMI (r = −0.569, *p* < 0.001) and adiponectin and glucose (r = −0.353, *p* = 0.018), also the positive correlations between leptin and HOMA2-IR (r = 0.466, *p* = 0.001) and IL-13 and GGT (r = 0.424, *p* = 0.004).

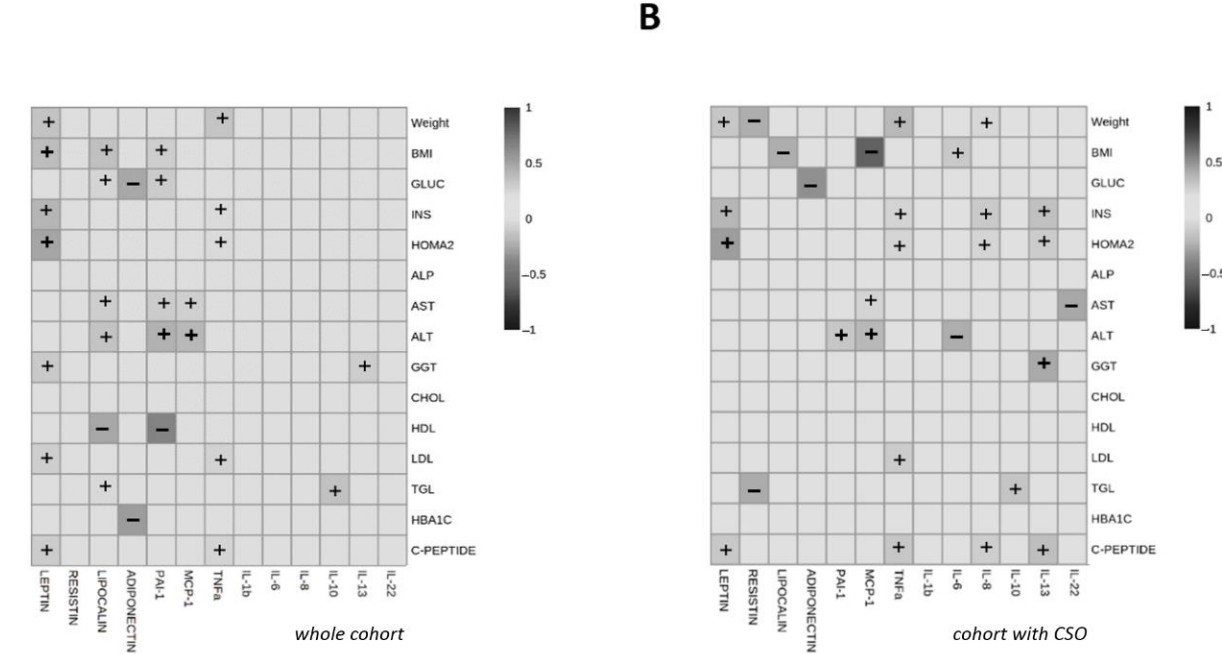

**Figure 4.** Correlation heat maps of associations between anthropometric and biochemical variables, interleukins and microbiota-derived metabolites. Pearson r correlations between anthropometric/biochemical variables and interleukins in the whole cohort of study (**A**) or in the cohort with clinically severe obesity (**B**). Positive correlations were marked with a (+) in a darker box; negative correlations were marked with a (−) in a darker box. Non-significant correlations were not marked. The darker the box, the more significant the association (*p* < 0.05). CSO, clinically severe obesity; BMI, body mass index; GLUC, glucose; INS, insulin; HOMA2, the homeostasis model assessment 2 of insulin resistance; ALP, alkaline phosphatase; CHOL, cholesterol; HDL, high density lipoprotein-cholesterol; LDL, low density lipoprotein-cholesterol; TGL, triglycerides; HbA1c, glycosylated hemoglobin A1c; PAI-1, plasminogen activator inhibitor-1; MCP-1, monocyte chemo attractant protein 1; TNF-α, tumor necrosis factor alpha; IL, interleukin.

*3.4. Correlations of Gut Microbiota Derived-Metabolites Levels with Anthropometrical Measures and with Metabolic Parameters Measures and with Metabolic Parameters*

At first, we evaluated the correlations between anthropometrical measures and metabolic parameters and gut microbiota derived-metabolites in the whole cohort (Figure 5A). We could highlight the positive associations between weight and TCA (r = 0.508, *p* < 0.001), insulin and GCA (r = 0.571, *p* < 0.001), GDCA (r = 0.522, *p* < 0.001), TCA (r = 0.664, *p* < 0.001), TCDCA (r = 0.689, *p* < 0.001) or TDCA (r = 0.652, *p* < 0.001); between GGT or c-peptide and TCDCA (r = 0.514, *p* < 0.001; r = 0.583, *p* < 0.001 respectively). When we studied only the cohort of patients with CSO (Figure 5B), we found positive correlations between weight and GCA (r = 0.585, *p* < 0.001) or TCA (r = 0.544, *p* < 0.001), insulin and DCA (r = 0.589, *p* < 0.001), GCA (r = 0.724, *p* < 0.001), GDCA (r = 0.647, *p* < 0.001), TCA (r = 0.687, *p* < 0.001), TCDCA (r = 0.769, *p* < 0.001), or TDCA (r = 0.696, *p* < 0.001). We also find relevant correlations between GGT and DCA (r = 0.508, *p* < 0.001), GCA (r = 0.528, *p* < 0.001), or TCDCA (r = 0.584, *p* < 0.001), and c-peptide and TCDCA (r = 0.668, *p* < 0.001) or GLCA (r = 0.522, *p* < 0.001).

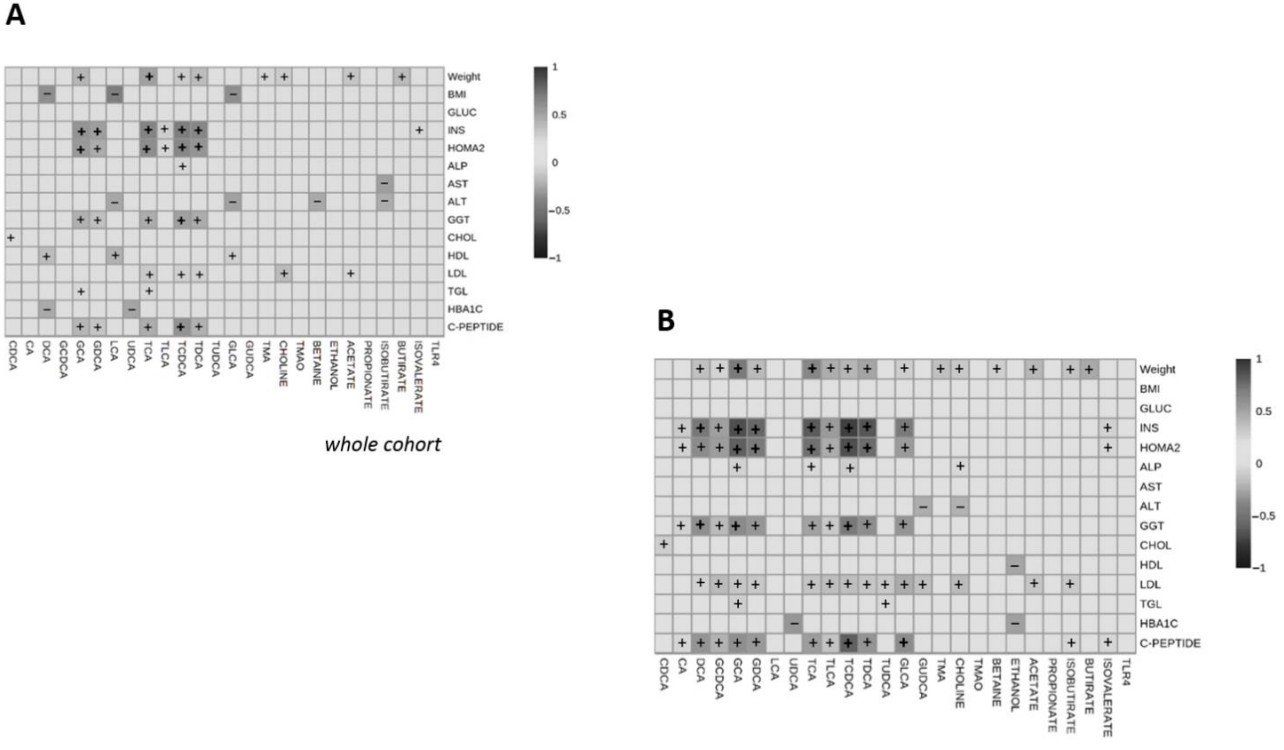

**Figure 5.** Pearson r correlations between anthropometric/biochemical variables and microbiota-derived metabolites in the whole cohort of study (**A**) or in the cohort with clinically severe obesity (**B**). Positive correlations were marked with a (+) in a darker box; negative correlations were marked with a (−) in a darker box. Non-significant correlations were not marked. The darker the box, the more significant association ($p < 0.05$). CSO, clinically severe obesity; BMI, body mass index; GLUC, glucose; INS, insulin; HOMA2, the homeostasis model assessment 2 of insulin resistance; ALP, alkaline phosphatase; CHOL, cholesterol; HDL, high density lipoprotein-cholesterol; LDL, low density lipoprotein-cholesterol; TGL, triglycerides; HbA1c, glycosylated hemoglobin A1c; CDCA, chenodeoxycholic acid; CA, cholic acid; DCA, deoxycholic acid; GCDCA, glycochenodeoxycholic acid; GCA, glycocholic acid; GDCA, glycodeoxycholic acid; LCA, lithocholic acid; UDCA, ursodeoxycholic acid; TCA, taurocholic acid; TLCA, taurolithocholic acid; TCDCA, taurochenodeoxycholic acid; TDCA, taurodeoxycholic acid; TUDCA, tauroursodeoxycholic acid; GLCA, glycolithocholic acid; GUDCA, glycoursodeoxycholic acid; TMA, trimethylamine; TMAO, trimethylamine N-oxide; TLR-4, Toll-like receptor-4.

*3.5. Correlations of Circulating Interleukin and Other Adipocytokine Levels with Gut Microbiota Derived-Metabolites Levels*

At the end, we wanted to analyze the association between circulating interleukin levels and gut microbiota derived-metabolites, first in the whole cohort (Figure 6A). To highlight, the positive correlations between TNF-$\alpha$ and choline (r = 0.601, $p < 0.001$) and between IL-22 and TMAO (r = 0.496, $p < 0.001$). Then, we analyzed these correlations in the cohort with CSO (Figure 6B). We found interesting positive associations regarding leptin and TCDCA (r = 0.425, $p = 0.004$), TNF-$\alpha$ and choline (r = 0.657, $p < 0.001$), TUDCA (r = 0.468, $p = 0.001$), or TLR4 (r = 0.557, $p < 0.001$); and IL-22 and TMA (r = 0.524, $p < 0.001$) or TMAO (r = 0.456, $p = 0.001$). We also can observe some interesting negative associations between resistin, adiponectin, or IL-10 and some microbiota-derived metabolites.

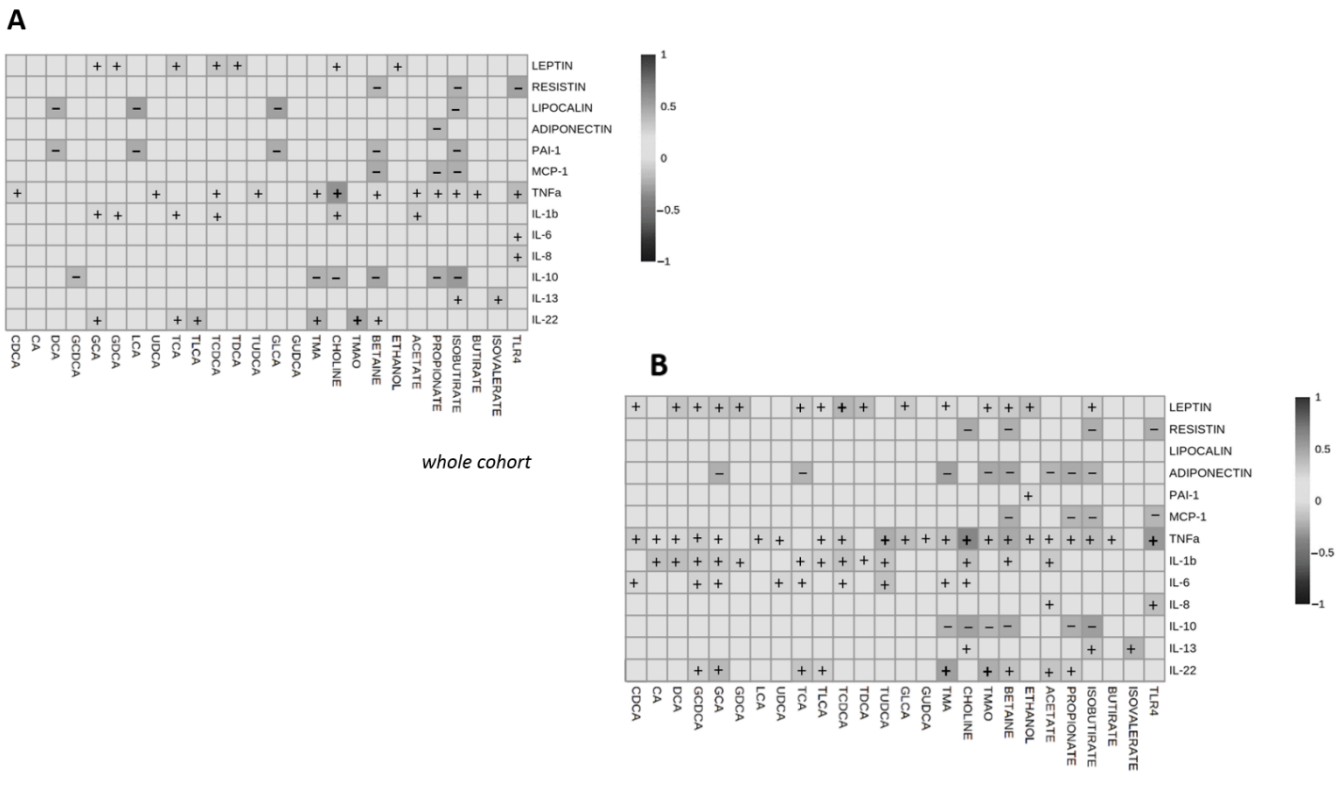

**Figure 6.** Pearson r correlations between interleukins and microbiota-derived metabolites in the whole cohort of study (**A**) or in the cohort with clinically severe obesity (**B**). Positive correlations were marked with a (+) in a darker box; negative correlations were marked with a (−) in a darker box. Non-significant correlations were not marked. The darker the box, the more significant association ($p < 0.05$). CSO, clinically severe obesity; PAI-1, plasminogen activator inhibitor-1; MCP-1, monocyte chemo attractant protein 1; TNF-α, tumor necrosis factor alpha; IL, interleukin; CDCA, chenodeoxycholic acid; CA, cholic acid; DCA, deoxycholic acid; GCDCA, glycochenodeoxycholic acid; GCA, glycocholic acid; GDCA, glycodeoxycholic acid; LCA, lithocholic acid; UDCA, ursodeoxycholic acid; TCA, taurocholic acid; TLCA, taurolithocholic acid; TCDCA, taurochenodeoxycholic acid; TDCA, taurodeoxycholic acid; TUDCA, tauroursodeoxycholic acid; GLCA, glycolithocholic acid; GUDCA, glycoursodeoxycholic acid; TMA, trimethylamine; TMAO, trimethylamine N-oxide; TLR-4, Toll-like receptor-4.

### 3.6. Predictive Value of Metabolic Parameters, the Preoperative Levels of Adipocytokines/ Interleukins and Gut Microbiota-Derived Metabolites on the Changes of BMI and Metabolic Factors

Prediction of change of BMI achieved post-surgery: To create a predictive algorithm, we studied the potential contribution of metabolic parameters, all the adipocytokines and gut microbiota-derived metabolites analyzed preoperatively on BMI reduction (BMI reduction = (Initial BMI − final BMI)/initial BMI × 100). Any adipocytokines/ interleukins or any microbiota-derived metabolites correlated with the BMI reduction.

### 3.7. Evaluation of the Change of the Glycemic Control Reached Postoperatively

Analysis of metabolic factors: HbA1c levels and HOMA2-IR. We evaluated the relationship between parameters of glycemic control postoperatively (HbA1c reduction = (Initial HbA1c − Final HbA1c)/Initial HbA1c × 100; HOMA2-IR reduction = (Initial HOMA2-IR − Final HOMA2-IR)/Initial HOMA2-IR × 100) and metabolic parameters and the adipocytokines/interleukins and microbiota-derived metabolites levels preoperatively. We found that HbA1c reduction after surgery were related positively to AST, ALT, and

lipocalin (r = 0.491, *p* = 0.006; r = 0.575, *p* = 0.001; r = 0.359, *p* = 0.043) and negatively with adiponectin (r = −0.416, *p* = 0.018), as shown Figure 7. Moreover, HOMA2-IR reduction after surgery was related positively to IL-1 β levels (r = 0.430, *p* = 0.032) and negatively to resistin levels preoperatively (r = −0.518, *p* = 0.008). A multiple linear regression analysis with backward variable selection was performed to find efficient predictors of HbA1c and HOMA2-IR reduction after bariatric surgery. We found that baseline lipocalin predicted HbA1c reduction (B coefficient = 0.443, *p* = 0.015) and baseline resistin levels predicted HOMA2-IR decrease (B coefficient = 0.518, *p* = 0.008).

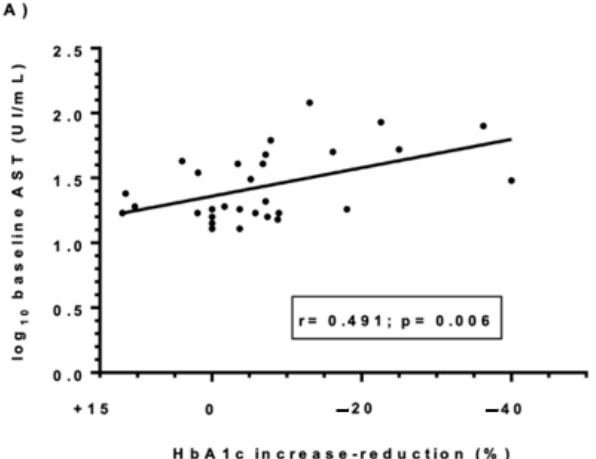

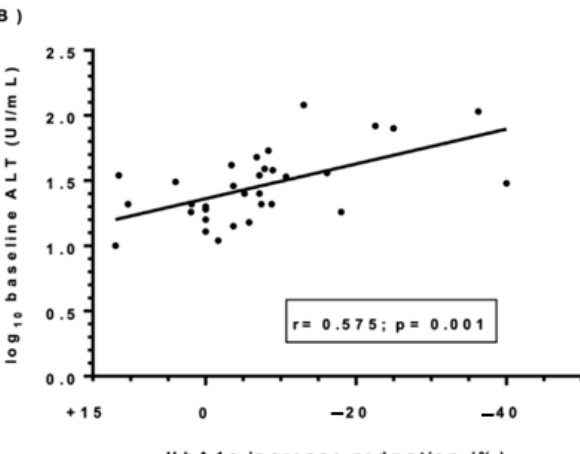

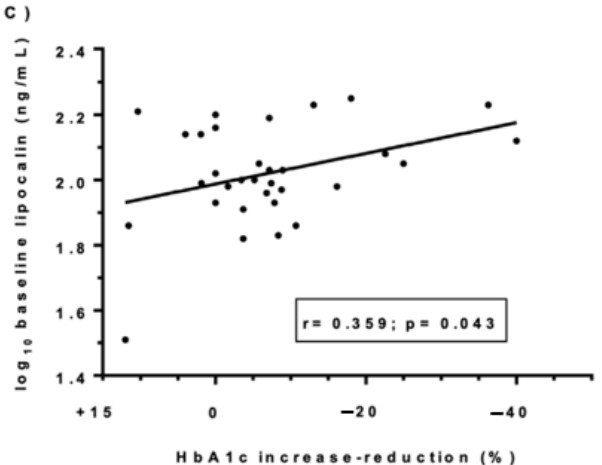

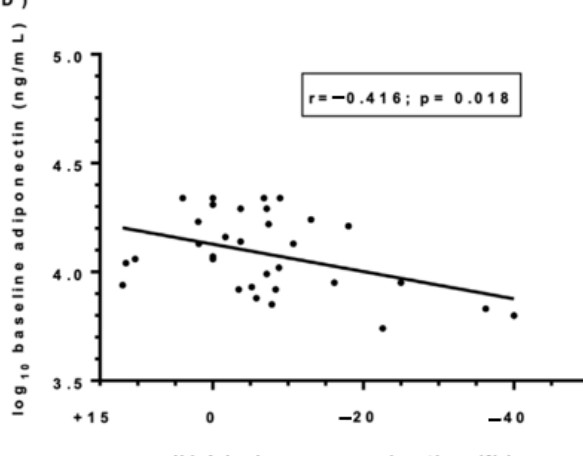

**Figure 7.** *Cont.*

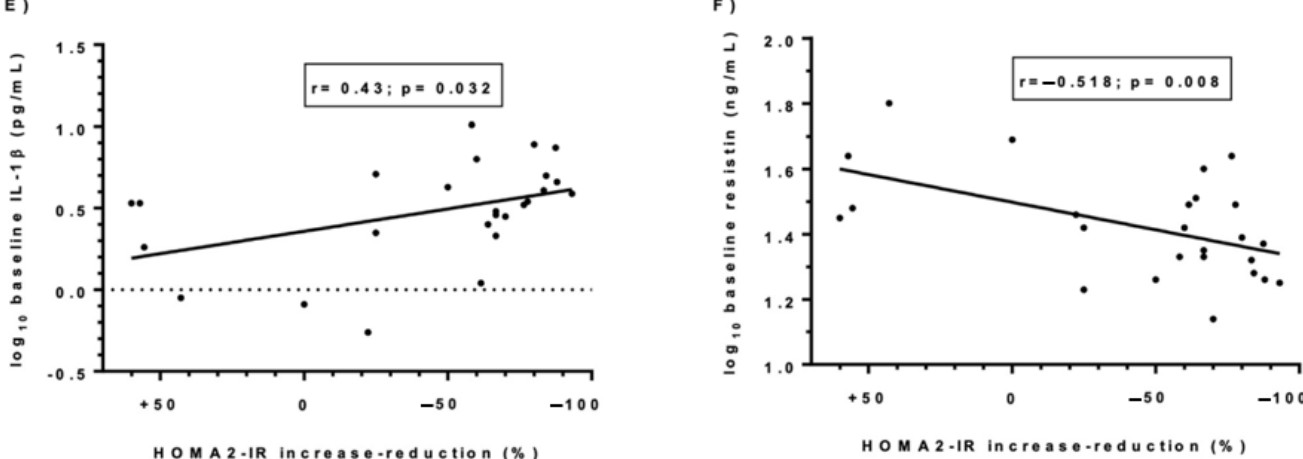

**Figure 7.** Spearman correlation analysis representation between the log10 baseline levels of the studied variables (AST (**A**), ALT (**B**), lipocalin (**C**), adiponectin (**D**), IL-1β (**E**), and resistin (**F**), respectively) and the glycemic control reached postoperatively (HbA1c or HOMA2-IR variability). HbA1c, glycosylated hemoglobin A1c; HOMA2, the homeostasis model assessment 2 of insulin resistance.; AST, aspartate aminotransferase; ALT, alanine aminotransferase; IL, interleukin. r means the rho of Spearman and *p*-value < 0.05 was considered statistically significant.

### 3.8. Evaluation of the Lipid Profile Postoperatively

Regarding lipid profile, at 12 months after surgery, LDL-C reduction (LDL-C reduction = (Initial LDL-C − Final LDL-C)/Initial LDL-C × 100) was related to AST, ALT, MCP-1, and propionate (r = −0.415, *p* = 0.016; r = −0.356, *p* = 0.036; r = −0.375, *p* = 0.029; r = 0.511, *p* = 0.008), as shown Figure 8. Any adipocytokines/interleukins or any microbiota-derived metabolites correlated with the HDL-C increase (HDL-C increase = (Final HDL-C − Initial HDL-C)/Initial HDL-C × 100). However, triglyceride reduction (triglyceride reduction = (Initial triglyceride − Final triglyceride)/Initial triglyceride × 100) was related to GGT, ALP, resistin, IL-22 and GCA (r = 0.330, *p* = 0.046; r = 0.393, *p* = 0.020; r = −0.368, *p* = 0.025; r = 0.339, *p* = 0.043; r = 0.375, *p* = 0.049).

When we developed a model to predict lipid profile 12 months after surgery, we only found that baseline propionate predicted LDL-C reduction (B coefficient = 0.531, *p* = 0.016).

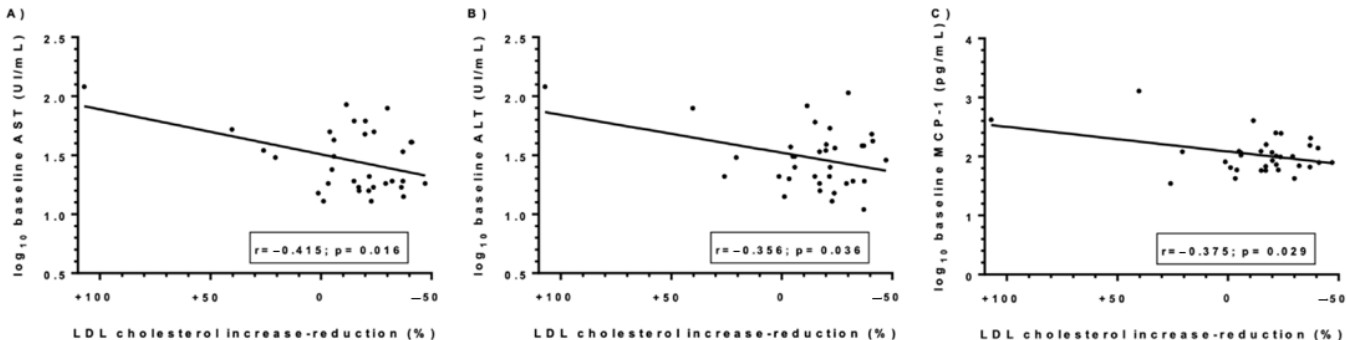

**Figure 8.** *Cont.*

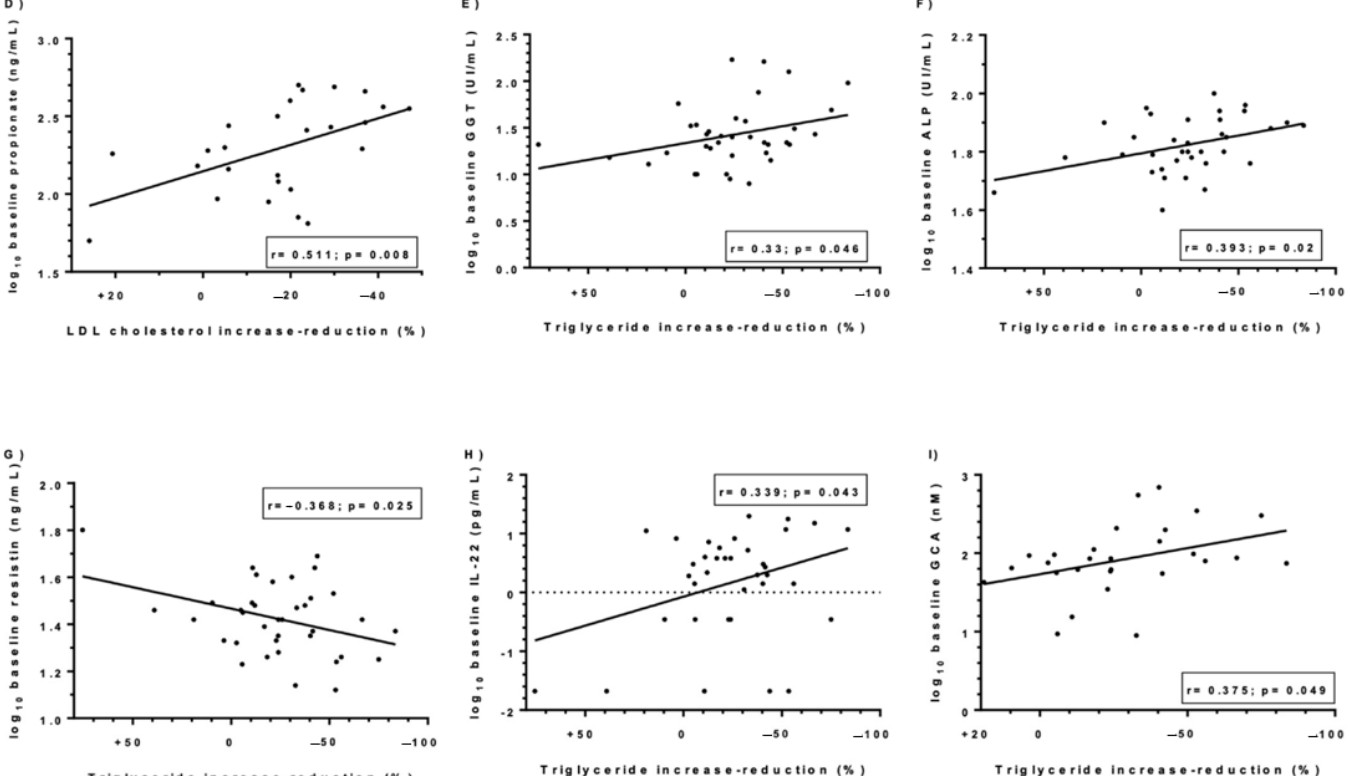

**Figure 8.** Spearman correlation analysis representation between the log10 baseline levels of the studied variables (AST (**A**), ALT (**B**), MCP-1 (**C**), propionate (**D**), GGT (**E**), ALP (**F**), resistin (**G**), IL-22 (**H**), and GCA (**I**), respectively) and the postoperatively lipid profile (LDL-C or triglyceride variability). LDL-C, low density lipoprotein-cholesterol; AST, aspartate aminotransferase; ALT, alanine aminotransferase; MCP-1, monocyte chemo attractant protein 1; GGT, gamma-glutamyltransferase; ALP, alkaline phosphatase; IL, interleukin; GCA, glycocholic acid. r means the rho of Spearman and *p*-value < 0.05 was considered statistically significant.

## 4. Discussion

In the present study, we corroborated the relevant weight loss and the improvement in metabolic factors after surgery in a cohort of patients made up of women presenting CSO. Moreover, the post-surgery levels of leptin, PAI-1, resistin, lipocalin, TNF-α, IL-13, IL-10, IL-8, and IL-1β were lower, whereas adiponectin levels were higher than their baseline levels. We also found that preoperative interleukin, other adipocytokines and gut microbiota-derived metabolite levels could be useful to predict changes in metabolic factors at 12 months postoperatively.

Regarding the improvement in BMI and metabolic factors after surgery, our results are similar to those described previously [2–4]. With regard to the changes in interleukin and other adipocytokine levels postoperatively in this small cohort of women, our findings are in line with those of other authors who described that bariatric surgery is linked to a reduction in specific adipocytokines, like leptin, chemerin, and PAI-1, meanwhile adiponectin levels are increased [21,22]. Similarly, the same authors (Askarpour, et al.), in a recent meta-analysis of clinical trials, suggested that bariatric surgeries might cause an important reduction in the levels of some inflammatory markers, such as C-reactive protein (CRP), IL-6, and TNF-α [23]. Our results agree with the fact that bariatric surgery is a technique than can ameliorate the inflammatory state in subjects with obesity given the increase in fat mass functionality and the decrease of some pro-inflammatory cytokine levels.

In our study, we also investigated baseline circulating concentrations of intestinal microbiota-derived metabolites in accordance to the presence or absence of obesity, and we found relevant differences in circulating choline, TMA, betaine, SCFAs, and some primary and secondary BA levels between women with CSO and women with NW. Aron-Wisnewsky et al. reported that the most part of gut microbiota alterations in CSO include a reduction in microbial gene richness and related-functional pathways linked with metabolic deterioration. They also described that after bariatric surgery, enterotype modification occurs, but most patients still have very low microbial gene richness [24]. In a previous study, we described similar findings [25], according to other authors [26,27].

Moreover, we described several associations between circulating interleukin and other adipocytokine levels, and gut microbiota-derived metabolite levels. It is well-known that intestinal microbiota-derived metabolites promote low-grade chronic inflammation of tissue-resident macrophages and are involved in diseases such as obesity, T2DM, metabolic syndrome, or cancer [28]. In this sense, most immune cells and membrane or intracellular receptors named "pattern recognition receptors" expressed on epithelial layer act as sensors of bacterial and cellular-derived products, pathogen-associated molecular patterns (PAMPs), and damage-associated molecular patterns (DAMPs) [29]. PAMPs and DAMPs such as LPS are recognized by members of the TLR family and nuclear oligomerization domain-like receptors of the NOD/NLR family. Extracellular and intracellular complexes formed by DAMPs and PAMPs and NOD/NLR receptors compose inflammasomes. Once the inflammasomes are activated, there is a production and release of interleukins (1β and IL-18). These cytokines promote the production of other pro-inflammatory cytokines, including TNF-$\alpha$, IL-6, IL-17, IL-22, and IL-23, and several active chemical inflammatory mediators [29]. It is also important to note the change in IL-10 levels at 12 months after surgical intervention, as this IL would indicate the well-known anti-inflammatory role of the procedure [30].

Then, we analyzed the predictive value of the preoperative levels of these molecules on weight loss and metabolic factor changes. An important finding of our work is that the baseline lipocalin levels are related to HbA1c reduction after surgery, and HOMA2-IR variability is associated with baseline resistin levels. Additionally, the preoperative levels of propionate are related to LDL-C variability post-surgery in this cohort. In this case, determining the levels of lipocalin, resistin and propionate prior to surgery may be useful for physicians to evaluate the metabolic impact that surgery would have on these patients. Although this information will not probably modify the surgery decision, it would be beneficial to improve baseline patients' metabolic conditions (personalized diet, probiotic intake, etc.) to ensure a greater long-term metabolic benefit. This could increase the overall effectiveness of the technique and decrease the percentage of patients who do not achieve the benefits it entails.

Regarding T2DM remission, some studies have been performed to identify some predictive pre-surgical factors. In a meta-analysis including 1753 bariatric surgery subjects, younger patients, short diabetes duration, better glycaemia control, and better β-cell function were more prone to reach T2DM remission after bariatric surgery [10]. Moreover, some predictive remission scores have been developed that include these factors (ABCD, DiaRem, AdDiaRem, and DiaBetter) [15,31–33]. In addition, a recent pilot study assessed genetic predisposition risk scores in T2DM and non-diabetic patients to predict a better response to bariatric surgery in terms of either weight loss or diabetes remission through a DNA study of saliva samples [34]. Additionally, gut hormones and succinate levels have also been studied as predictors of T2DM remission [11,12]. The visceral adiposity index has even been used to predict remission of T2DM [35] or the gut microbiota profile [36].

With regard to the prediction of the lipid profile postoperatively, it was reported that preoperative HDL-C and the type of surgery assessed are predictors of the increase in HDL-C levels in this cohort [37]. However, in our study, only baseline propionate levels were related to LDL-C prediction in our cohort.

The present study has some limitations. These preliminary results were obtained in a homogeneous cohort of women with CSO after a follow-up period of 12 months. However, the predictive power of interleukins and other adipocytokines and gut microbiota-related metabolites levels is weak, and it needs to be assessed in a larger independent cohort including both sexes, and perhaps a longer follow-up period.

## 5. Conclusions

To conclude, the detection of some adipocytokines (lipocalin and resistin) and a short-chain fatty acid (propionate) could be useful to predict the improvement of metabolic changes after bariatric surgery. Although the surgical decision will not be modified based on this prediction, it would be necessary to modify the medical treatment to improve baseline metabolic conditions. This study provides preliminary results that should be validated in other cohorts and new knowledge is needed.

**Author Contributions:** Conceptualization, T.A. and C.R.; methodology, M.L.-D., L.B. and C.A.; software, T.A. and M.L.-D.; validation, J.R., D.R., A.A. and M.P.; formal analysis, T.A., M.L.-D., J.R. and L.B.; investigation, T.A. and J.R.; resources, F.S. and D.D.C.; data curation, T.A., M.L.-D., L.B. and C.A.; writing—original draft preparation, T.A.; writing—review and editing, L.B.; visualization, T.A.; supervision, T.A.; project administration, T.A.; funding acquisition, T.A. and C.R. All authors have read and agreed to the published version of the manuscript.

**Funding:** This research was funded by Instituto de Salud Carlos III grant number PI16/00498 (to T.A.) (Co-funded by European Regional Development Fund "A way to make Europe"), by funds from Agència de Gestió d'Ajuts Universitaris de Recerca (AGAUR 2017 SGR 357 to C.R.) and the Grup de Recerca en Medicina Aplicada URV (2016PFR-URV-B2-72 to C.R.).

**Institutional Review Board Statement:** The study was conducted in accordance with the Declaration of Helsinki, and approved by the Institutional Review Board of Institut d'Investigació Sanitària Pere Virgili (CEIm: 161C/2016; 26 April 2016).

**Informed Consent Statement:** Written informed consent has been obtained from the patient(s) to publish this paper.

**Conflicts of Interest:** The authors declare no conflict of interest. The funders had no role in the design of the study; in the collection, analyses, or interpretation of data; in the writing of the manuscript, or in the decision to publish the results.

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
