# Peer review of "Lipocalin, Resistin and Gut Microbiota-Derived Propionate Could Be Used to Predict Metabolic Bariatric Surgery Selected Outcomes"

_processes, doi:10.3390/pr10010143_

Round 1

Reviewer 1 Report

The Authors studied the predictive value of adipocytokines and microbiota-derived metabolites for the metabolic improvement following surgery in women with clinically severe obesity (CSO). For this prospective study they recruited 44 women with CSO and 21 women with normal weight, as control group. They analyzed circulating levels of some interleukins, other adipocytokines and some microbiota-derived metabolites at the moment of surgery and 12 months later. The results showed that glucose, insulin, HbA1c, LDL-C and triglycerides levels were reduced postoperatively; while HDL-C levels increased. Twelve months later, leptin, resistin, lipocalin, PAI-1, TNF-α, and IL-1β levels were lower than baseline levels, meanwhile adiponectin, IL-8 and IL-10 levels were increased. The Authors suggest that the best predictor of HbA1c reduction post-surgery is baseline lipocalin; meanwhile for HOMA2 (insulin resistance) is the baseline resistin, and for LDL-C decrease is propionate. They conclude that a preoperative panel for the detection of lipocalin, resistin and propionate levels may be used to predict the metabolic success following bariatric surgery. The study was well planned and carried out, but major concerns derive from data presentation and discussion. The Authors had to present a rational framework for the prediction they would like to suggest in these cases of bariatric surgery.                      

Reviewer 2 Report

The manuscript prepared by Teresa Auguet et al., entitled ''Lipocalin, resistin and propionate in a preoperative panel to predict metabolic bariatric surgery selected outcomes'' is a research article that provides important data regarding the role of bariatric surgery in the balance of circulating adipokines and gut microbiota-related metabolites factors, in a cohort of severe obesity women.

In obese patients, recent reports revealed that bariatric surgery is associated with a reduction in specific adipokines including leptin, chemerin, and PAI-1, whereas adiponectin is raised, adaptations that could be indicative of fat mass and function.

To improve the manuscript we recommend:

  • the title does not reflect the data presented in the main text, therefore the authors should reformulate it to include the gut microbiota-derived metabolites.
  • please mention the complete name of all abbreviations in brackets at the beginning (see lines 26-29 or BMI in lines 86-87).
  • In Materials and Methods (lines 136-139) the authors should add an extension of data presented regarding the methods used in this research.
  • In sections 3.3, 3.4, 3.5, 3.7, 3.8 the results should be associated with figures that could be valuable to highlight these sections and the data presented by the authors.
  • kindly complete the Discussion section related to the relationship between bariatric surgery and reduction in the levels of some inflammatory markers. It is abrupt and incomplete.
  • The English language and typing errors should be checked.

As minor recommendations:

  • what means RYGB in line 46?
  • In line 19 - microbiota-derived metabolites should be replaced with gut/intestinal microbiota-derived metabolites.

Reviewer 3 Report

The predictive potential of this small study is limited. While the authors acknowledge this in their conclusions, the mention of predictive value of the results observed should be minimized. 

Round 2

Reviewer 2 Report

The authors have addressed the reviewer's comments well. The quality of the manuscript has greatly improved with more details and clarifications.

Author Response

Dear reviewer, we appreciate you implication in the review process of this manuscript, in this sense we have had the opportunity to improve our content. Thank you so much.